# Prediction of hydrological and water quality data based on granular-ball rough set and k-nearest neighbor analysis

**Limei Dong[1], Xinyu Zuo[1], Yiping Xiong[2]** *

**1** Upper Changjiang River Bureau of Hydrological and Water Resources Survey, Chongqing, China, **2** College of Computer Science and Technology, Chongqing University of Posts and Telecommunications, Chongqing, China

* 870255728@qq.com

## Abstract

Hydrological and water quality datasets usually encompass a large number of characteristic variables, but not all of these significantly influence analytical outcomes. Therefore, by wisely selecting feature variables with rich information content and removing redundant features, it not only can the analysis efficiency be improved, but the model complexity can also be simplified. This paper considers introducing the granular-ball rough set algorithm for feature variable selection and combining it with the k-nearest neighbor method and back propagation network to analyze hydrological and water quality data, thus promoting overall and fused inspection. The results of hydrological water quality data analysis show that the proposed method produces better results compared to using a standalone k-nearest neighbor regressor.

## 1. Introduction

Algal blooms are a natural ecological phenomenon in water bodies, but only caused by algae [1]. Water bloom (or harmful algal blooms) is a phenomenon in which a large number of algae grow and reproduce or accumulate and reach a certain concentration when the water body is rich in nutrients and has suitable environmental conditions such as temperature, light, climate and suitable hydrological conditions that are conducive to the growth or aggregation of algae [2]. Algae is one of the common biological components in water, but excessive growth of algae will bring problems. Excessive nutrients, especially phosphorus and nitrogen, are the main causes of algal blooms [3]. In freshwater bodies, the main dominant algae species causing blooms are Aphanizomenon, Microalgae, Anabaena, Empty Chlorella or Cryptococcus. The colors of different algae-forming blooms are different due to different algae species. The main colors are green, red and brown. The species and quantity distribution of algae have obvious regional and seasonal characteristics [4, 5]. This is because the composition of phytoplankton in water is closely related to the physical and chemical properties of water. Different algae adapt to different physical and chemical properties of water. For example, warm-loving species prefer water with higher water temperature, and acid-resistant species prefer water with lower

confidential data. Contact information:
syzuoxy@cjh.com.cn.

**Funding:** The author(s) received no specific funding for this work.

**Competing interests:** We declare that we have no financial and personal relationships with other people or organizations that can inappropriately influence our work, there is no professional or other personal interest of any nature or kind in any product, service and/or company that could be construed as influencing the position presented in, or the review of, the manuscript entitled. This does not alter our adherence to PLOS ONE policies on sharing data and materials.

pH values. Therefore, with the different external conditions, the algae species that bloom are different [6].

At present, the process and mechanism of freshwater water blooms are not very clear. Most experts believe that algal bloom is a short-term massive reproduction and growth of algae, and the occurrence process of algal bloom is divided into the incubation period (algal bloom begins to reproduce rapidly), peak period (the number of algal bloom reaches the peak) and dissipation period (the number of algal bloom decreased significantly). Recently, experts from the Chinese Academy of Sciences have proposed that the "outbreak" of cyanobacterial blooms is only the apparent phenomenon caused by the floating and accumulation of algae in a certain period of time, that is, the change of spatial position, rather than the great change of biomass [7]. However, the author believes that the occurrence of water bloom is the result of the interaction of the above two processes. According to statistics, many lakes [8–13], reservoirs [14–19] and rivers [20, 21] are currently reported to have algal blooms. Therefore, the problem of water bloom is the main problem that has been or will be faced by most water bodies in China. When algal blooms occur, the concentration of microcystins in water will increase significantly. There are three main chemical structures of microcystins detected in freshwater blooms: cyclic peptides, alkaloids and lipopolysaccharides [22]. One of the most important microcystins is microcystins. Microcystins are a group of toxic compounds that are currently studied due to their high toxicity and wide distribution. They are secondary metabolites produced by some strains or species of Microcystis, Anabaena, Oscillatoria and Nostoc in cyanobacteria. Not only the problem of freshwater water blooms is serious, but also the algae blooms in the main coastal waters (often referred to as red tides) can not be ignored. Red tides have been reported in the main coastal waters of China, such as the Bohai Sea [23, 24], the Yellow Sea [25, 26], the East China Sea [27–29], the Taiwan Strait [30, 31], and the South China Sea [32–34].

The analysis, modeling and prediction of hydrology and water quality are of positive significance for the prediction and prevention of water bloom. In 1983, Okada and Aiba [35] proposed a model to simulate the vertical spatial distribution of Microcystis biomass in calm water. The model simulates the emergence and disappearance of Microcystis blooms on the surface of eutrophic water by determining the vertical and spatial distribution of photosynthesis rate and buoyancy. The model does not consider the competition between freshwater plankton and the food chain. In 1984, Reynolds [36] constructed a nomogram to explain the biomass changes of algal populations using algal population biomass data related to seasonal and lake nutritional status (in this study, the biomass data of 49 different algal populations in 12 temperate lakes were used to construct a nomogram). SK Jayaraman et al. (2015) [37] used a multi-level direct search optimization technique with steady-state convergence criteria to predict the growth of microalgae. However, this method studied a closed environment and was not suitable for hydrological and water quality analysis. The study of Pyo J C et al. (2021) [38] showed a pan-Arctic result obtained by combining a new submarine algae sea ice model (SIMBA) with a three-dimensional sea ice-ocean model. The model is evaluated based on data collected during an onboard activity in the mid-eastern Arctic in the summer of 2012. Algae blooms are caused by light and show latitude dependence. Snow and ice also play a key role in the growth of ice algae. These studies can be used for auxiliary analysis of freshwater algae research.

The inductive modeling method uses regression analysis and/or correlation analysis to extract knowledge and patterns from a large amount of historical data to construct the overall model of the system and uses the constructed overall model to predict system behavior (fewer components to explain system behavior). Sakamoto (1966) [39] and later Dillon & Rigler (1975) [40] constructed the first empirical data model, which used Total Phosphorous data to

predict the amount of chlorophyll (chlorophyll-a) in lakes (chlorophyll quantity can be regarded as an estimate of total algae biomass). In 1984, Whitehead & Horberger [41] applied the time series method to construct a model to predict the change of chlorophyll content in the Thames River. In 1994, Recknagel et al. [42] also used the fuzzy logic method to model and predict the change of chlorophyll content in lakes and reservoirs. Many scholars have used regression analysis to study the relationship between environmental variables and red tides [43–48]. In addition, fuzzy logic, as a mathematical method commonly used for modeling and prediction of complex systems [49, 50], has also been used for modeling and predicting algal blooms [50–52]. In recent years, machine learning-based methods have been used to predict and evaluate the content of various algae [51, 52]. Among them, k-nearest neighbor (kNN) is one of the most basic and simple machine learning algorithms [53, 54]. It is also used for inter-annual remote sensing monitoring of eelgrass beds [55], algae toxicity identification [56] and water quality assessment [57, 58].

With the rise of neural network applications, in the middle and late 1990s, some scholars began to apply neural networks to algae ecological modeling. Because the Back Propagation (BP) network algorithm is mature and easy to use, it is the most widely used in modeling and prediction research. Recknagel (1997) [59] used a three-layer feedforward neural network to construct a cascade-structured neural network model to model and predict the biomass of five algae species in Lake Kasumigaura, Japan. The results show that the inductive type of empirical model has good predictive ability for complex lake and marsh phenomena, but how to extract useful knowledge from the neural network weights to explain the occurrence mechanism of algal blooms is worthy of further research. In addition, Recknagel et al. [59] used BP neural network to conduct predictive modeling research on a large amount of historical data from Lake Kasumigaura and Lake Biwa in Japan, Lake Tuusulanjaervi in Finland, and Darling River in Australia. The prediction results show that the artificial neural network method can effectively predict complex nonlinear algal bloom phenomena in freshwater ecosystems under different environmental conditions. Although it can basically reflect the peaks and troughs of algal population biomass, the error between it and the actual observation value is relatively large. Wei et al. (2001) [60] also applied BP neural network modeling to algae ecological modeling in Kasumigaura Lake, Japan, and could well predict the occurrence time and intensity changes of Microcystis blooms. Maier, Dandy (1997) [61] and Maier et al. (1998) applied the BP network to the algae ecological modeling of rivers, using historical data of climate, hydrology, nutrient levels and other factors in the Murray River in Australia to predict Anabaena algae changes in biomass. BP neural network is still the mainstream of current algae ecological modeling, but most methods are also analyzed based on all characteristic attributes of the data, lacking correlation analysis between features.

Most existing methods for modeling and predicting algae-based biomass involve all attributes of the data in decision-making, and some redundant attributes may affect the result analysis. The heuristic attribute reduction algorithm based on rough set theory can effectively reduce the time complexity of high-dimensional problems [62–64]. It can also detect algae more efficiently and accurately. The latest research shows that granular-ball rough set (GBRS) [65] based on granular-ball computing [66] combines the robustness and adaptability of granular-ball computing. This method can handle discrete and continuous data stably and efficiently, and feature selection is even better than the state-of-the-art methods. Therefore, we consider combining GBRS and regression methods to analyze hydrological and water quality data. The detailed main contributions are as follows:

- This paper uses GBRS to remove redundant features in water quality data and select feature variables with rich information content for data analysis. And the Pearson correlation

coefficient method was used for feature correlation analysis and compared with the GBRS method.

- The results after feature selection are input into kNN and BP neural networks for regression prediction.

- The hydrology and water quality data analysis results show that combining the GBRS method with kNN to analyze hydrology and water quality data produces better results.

The rest of this paper is organized as follows. We introduce related works in Section 2. Experimental results and analysis are presented in Section 4. We present our conclusion and future work in Section 5.

## 2. Materials and methods

### 2.1 Granular-ball rough set

Human cognition possesses a cognitive mechanism known as "global priority" [67], which enables the processing of information input based on coarse-grained details, thus providing adaptive multi-granularity descriptive capabilities. Building upon this theory, Wang and Xia [66] proposed multi-granularity sphere computing by using the granular-ball to cover sample points. This approach replaces individual sample points with the granular-ball as inputs, significantly reducing the number of required training samples. Additionally, the coarse-grained nature of the granular-ball ensures that they are less susceptible to the influence of fine-grained sample points, thereby enhancing the algorithm's robustness. Each granular-ball $GB = \{x_i, i = 1, 2, \cdots, N\}$ can be expressed using only two features: its center $c$ and radius $r$, applicable across any dimension. The expression is as follows [66]:

$$C = \frac{1}{N}\sum_{i=1}^{N}x_i, r = \frac{1}{N}\sum_{i=1}^{N}\|x_i - C\|, \tag{1}$$

where $N$ denotes the number of samples in the granular-ball. The size of the radius in (1) indicates the different granularity of balls. Larger radii result in fewer granular-balls, indicating a coarser level of granularity. More efficient granular-ball computation contributes to improved algorithm robustness. Currently, this method has been successfully employed in the field of rough sets. While the Pawlawk rough set utilizes equivalence classes for knowledge representation, it cannot handle continuous data. Conversely, neighborhood rough sets can address continuous data, but encounter the challenge of "heterogeneous transmission," hindering knowledge representation. To overcome these limitations, Xia et al. [65] introduced granular-balls into rough set theory, proposing granular-ball rough sets (GBRS). This framework allows for the processing of continuous data while utilizing equivalence classes for knowledge representation. The specific models are defined and described as follows.

Rough set theory is a key component of granular computing. Pawlak rough set and neighborhood rough set are two important models within rough set theory. Pawlak rough set represents knowledge through equivalence classes but cannot handle continuous data. On the other hand, neighborhood rough set can handle continuous data but suffers from the "heterogeneous transmission" problem, which hinders knowledge representation. Granular-balls can simultaneously represent continuous data and perform equivalence class partitioning. Therefore, Xia et al. [65] introduced granular-ball into rough set theory and established the concept of granular-ball rough set, which is a unified learning model of Pawlak rough set and neighborhood rough set. Granular-ball rough set refers to the rough set based on granular-ball computing, the model is shown as follows.

**Definition 1.** [65] Let $U = \{x_1, x_2, \cdots, x_n\}$ is a non-empty finite set of real space. $\forall x_i \in U$ a granular-ball $GB_j$ is defined as:

$$GB_j = \{x | x \in U, \Delta(x, c_j) \leq r_j\}, \tag{2}$$

where $c_j$ and $r_j$ denotes the center of $GB_j$ and the radius of $GB_j$, respectively. Obviously, the larger the radius $r_j$ of the granular-ball, the coarser the granularity size and vice versa, the finer the granularity size.

**Definition 2.** [65] Let $\langle U, A, V, f \rangle$ be an information system, and $U$ is the set of objects. $A$ and $V$ denotes the set of all attributes and the values of attributes respectively, and $f$ denotes a mapping function that $f: U \times A \rightarrow V$. $\forall x, y \in U$ and $B \subseteq A$, the indiscernible granular-ball relation $INDGB(B)$ of the attribute subset $B$ is defined as:

$$INDGB(B) = \{(x, y) \in U^2 | f(x, a) = f(y, a) = GB, \forall a \in B\}, \tag{3}$$

where $a$ is an attribute of $B$, if $(x, y) \in INDGB(B)$, then $x$ and $y$ are indiscernible according to attribute set $B$, denoted as $x \sim y$. In granular-ball rough set, $INDGB(B)$ denotes an equivalence relation on $U$, which can create a partition of $U$, denoted as $U/GB(B)$. An element $[x]_{GB(B)} = \{y \in U | (x, y) \in INDGB(B)\}$ in $U/GB(B)$ is an equivalence class generated by granular-ball computing.

**Definition 3.** [65] Let $\langle U, A, V, f \rangle$ be an information system. For $\forall a \in B$, $GBR_B$ denotes a corresponding relation on $U$. $\forall X \in U$, the upper and lower approximation of x based on attribute set $B$ can be described as follows:

$$\overline{GBR_B}X = \cup\{[x]_B \in \mathcal{U}/GB(B) | [x]_{GB(B)} \cap X \neq \emptyset\}, \tag{4}$$

$$\underline{GBR_B}X = \cup\{[x]_B \in \mathcal{U}/GB(B) | [x]_{GB(B)} \subseteq X\}. \tag{5}$$

## 2.2 Pearson correlation coefficient

Pearson correlation coefficient is a simple method that can help understand the relationship between features and response variables. This method measures the linear correlation between variables. The value range of the result is [-1,1]. "-1" and "+1" indicates a complete negative correlation and a complete positive correlation respectively, and 0 indicates no linear correlation. The Pearson correlation coefficient is fast and easy to calculate and is often performed the first time after obtaining the data (after cleaning and feature extraction). Specifically defined as follows:

If two sets of data $X = \{X_1, X_2, \cdots, X_n\}$ and $Y = \{Y_1, Y_2, \cdots, Y_n\}$ are overall data (such as census results), then the overall mean is expressed as [68]

$$E(X) = \frac{\sum_{i=1}^{n} X_i}{n}, E(Y) = \frac{\sum_{i=1}^{n} Y_i}{n}. \tag{6}$$

Population covariance is

$$Cov(X, Y) = \frac{\sum_{i=1}^{n}(X_i - E(X))(Y_i - E(Y))}{n}. \tag{7}$$

Overall Pearson correlation coefficient can be obtained by

$$\rho_{XY} = \frac{Cov(X, Y)}{\sigma_X \sigma_Y} = \frac{\sum_{i=1}^{n} \frac{(X_i - E(X))}{\sigma_X} \frac{(Y_i - E(Y))}{\sigma_Y}}{n}. \tag{8}$$

$\sigma_X$(*sigma X*) is the standard deviation of *X*, where

$$\sigma_X = \sqrt{\frac{\sum_{i=1}^{n}(X_i - E(X))^2}{n}}, \sigma_Y = \sqrt{\frac{\sum_{i=1}^{n}(Y_i - E(Y))^2}{n}}. \tag{9}$$

It can be proved that $|\rho_{xy}| \leq 1$, if and only if $Y = aX + b$, $\rho_{XY} = \begin{cases} 1, a > 0 \\ -1, a < 0. \end{cases}$

Observing the formula of the overall Person correlation coefficient, we find that the Pearson correlation coefficient can be seen as eliminating the dimensional influence of the two variables, that is, the covariance after *X* and *Y* standardization. Therefore, we can use the Pearson correlation coefficient to measure the degree of linear correlation between the two variables. As shown in S1 Fig, the correlation between two variables *x* and *y* can be easily determined by drawing a scatter plot. Only when the following conditions are met, can we accurately use the Pearson correlation coefficient to judge: first, there is a linear relationship between the two variables, and they are continuous data; secondly, the overall distribution of the two variables is normal distribution, or near normal unimodal distribution; finally, the observations of the two variables are paired, and each pair of observations is independent of each other. By observing the graphics and verification conditions, the linear relationship between these two variables can be obtained.

## 2.3 kNN regressor

According to the given distance measurement method (generally using Euclidean distance), the *k* sample points closest to *x* are found in the training set $T = \{(x_i, y_i), i = 1, 2, \ldots, N\}$, and the set represented by these *k* sample points is recorded as $N_k(x)$. In $N_k(x)$, the category *y* is determined by the classification decision rule (such as majority decision) *x*, $y = \sum_{x_i \in N_k(x)} I(y_i, c_j), i = 1, 2, \ldots, N; j = 1, 2, \ldots, k$. In the above formula, *I* is an indicator function: $I_{x,y} = \begin{cases} 1, if x = y \\ -1, if x \neq y. \end{cases}$ The basic idea of the kNN regression algorithm is to select the nearest training samples in *k* Euclidean spaces according to the similarity of Euclidean distance between samples, and the regression value is the mean or weighted value of *k*-nearest neighbor samples. The specific steps of the kNN regression algorithm are as follows: (1) There are *n* training samples, which are expressed as $X = (X_1, X_2, \ldots, X_n)$, and each training sample can be expressed as $X_i = (x_{i1}, x_{i2}, \ldots, x_{id}, y_i), i \in n$. Then the Euclidean distance *D* between the training sample $X_i$ and the test sample $X_t = (x_{t1}, x_{t2}, \ldots, x_{td}, y_t), t \in n$ can be expressed as [69]

$$D(X_i, X_t) = \sqrt{\sum_{m=1}^{d}(x_{im} - x_{im})^2 + (y_i - y_t)^2}, \tag{10}$$

where $x_{im}$ and $x_{tm}$ are the observed values of the *m*th auxiliary variable (the variable used to build the model); $y_i$ and $y_t$ are the monitoring variables' observation value. (2) Calculate the Euclidean distance between all training samples and test samples according to (1) and find the first *K* nearest neighbor samples $X_j' = (x_{j1}', x_{j2}', \ldots, x_{jd}', y_j'), j \in k$. (3) Calculate the estimation value $\hat{y}_t$ of the monitoring variable for the test sample $X_t$.

$$\hat{y}_t = \frac{1}{k}\sum_{j=1}^{k} y_j'. \tag{11}$$

## 2.4 BP neural network

The BP neural network is a multi-layer feedforward network trained according to the error back propagation algorithm. It is one of the most widely used neural network models currently. It is a supervised learning algorithm with strong adaptive, self-learning and non-linear mapping capabilities. It can better solve the problems of small data, poor information and uncertainty, and is not restricted by non-linear models. The BP neural network can learn and store a large number of input-output pattern mapping relationships without revealing the mathematical equations describing the mapping relationship in advance. Its learning rule is to use the steepest descent method to continuously adjust the weights and thresholds of the network through backpropagation to minimize the sum of squared errors of the network. The multi-layer perceptron can obtain the final output $a^{(L)}$ of the network through layer-by-layer information transmission. The whole network can be viewed as a composite function $\emptyset(x : \omega, b)$. It takes the vector $x$ as the output of the first layer $a^{(0)}$. The output of the $L$ layer is used as the output of the whole function.

$$x = a^{(0)} \rightarrow z^{(1)} \rightarrow a^{(1)} \rightarrow z^{(2)} \rightarrow \cdots \rightarrow a^{(L-1)} \rightarrow z^{(L)} \rightarrow a^{(L)} = \emptyset(x : \omega, b), \qquad (12)$$

where $\omega$ and $b$ denote the connection weights and biases of all layers in the network. The error backpropagation algorithm corrects the $\omega$ and $b$ between each neuron by assigning the error to each hidden layer neuron so that the error signal tends to be minimized. This paper selects the multi-layer BP neural network model. Fig 1 is the algorithm flow chart of the BP neural network.

## 3. Method

In order to analyze the correlation between data features, we first combine the Pearson correlation coefficient and the kNN algorithm or BP neural network for analysis. The specific flow chart of the method is shown in Fig 2. Firstly, the correlation coefficients between the condition attributes and the label attributes are calculated, and then these coefficients are sorted. After deleting the attributes corresponding to the largest coefficients, the kNN algorithm or BP neural network is executed in the previous condition attributes, and the accuracy of the algorithm is calculated. Perform the above process in turn, and the final attribute set corresponding to the highest accuracy is the optimal attribute set after feature selection.

Although the Pearson correlation coefficient method can achieve feature selection, this method selects the optimal accuracy through kNN or BP neural network calculation based on the results of feature correlation analysis, which is relatively complicated. We considers introducing the GBRS method to directly select some data features, remove redundant attributes, and then process the data through regression methods. The flow chart of this method is shown in Fig 3. First, initialize the attribute set, add conditional attributes in sequence, and regenerate the granular-ball. If the sample size covered by granular-balls increases, retain the conditional attribute and repeat the above steps. Otherwise, output the conditional attribute set and

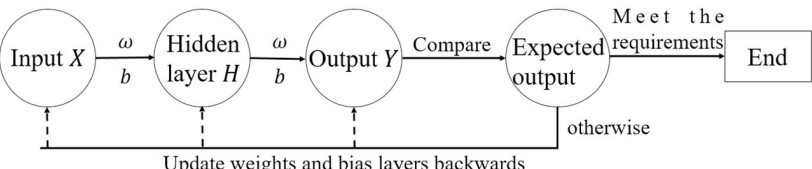

**Fig 1. The algorithm flow chart of BP neural network.**

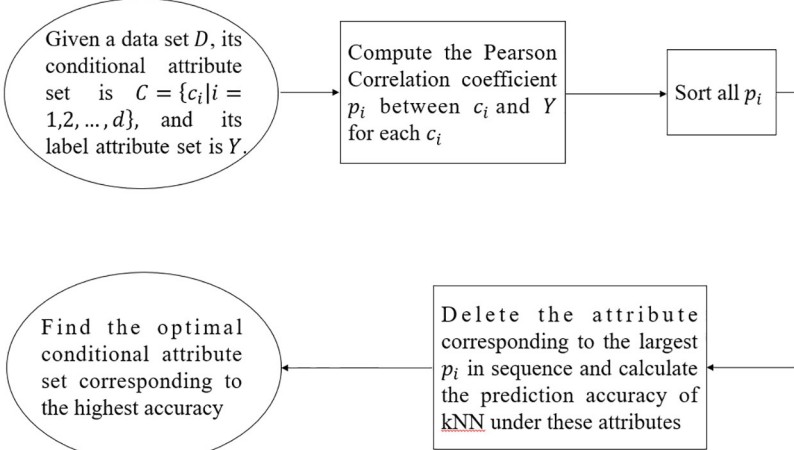

**Fig 2. Flow chart of prediction method based on pearson correlation coefficient and kNN regressor.**

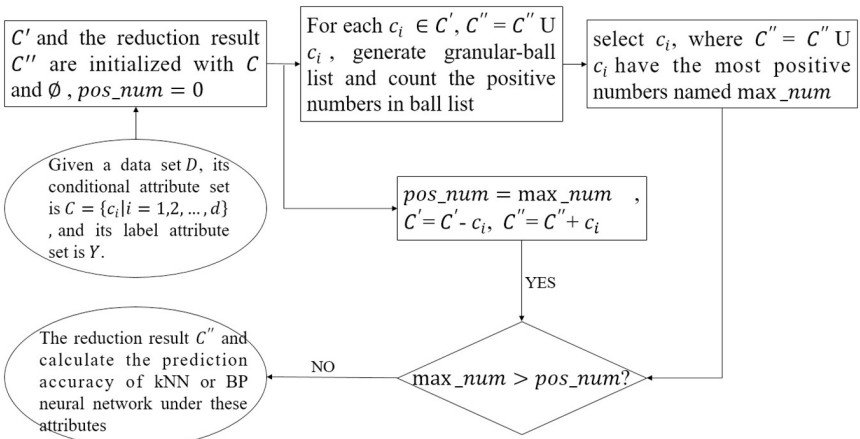

**Fig 3. Flow chart of prediction method based on GBRS and kNN regressor.**

calculate the prediction accuracy of kNN or BP neural network under these attributes. The GBRS can simultaneously represent both the Pawlak rough set and the neighborhood rough set, enabling it not only to be able to deal with continuous data but to use equivalence classes for knowledge representation as well. It combines the robustness and adaptability of the granular-ball computing, which enables the flexible fitting of different data distributions using the granular-balls with various radii. Thus, it prevents the propagation of heterogeneity caused by the overlap between the positive regional neighborhoods of different labels in the NRS. In addition, because the granular-ball set is used instead of points as the input, it can achieve higher efficiency than the feature selection on the basis of coarse-grained.

## 4. Results

### 4.1 Sample pretreatment

The water level and flow data used in this experiment are measured data from the hydrological station of the Hydrology and Water Resources Survey Bureau of the Upper Yangtze River, and

the water quality and ecological data are obtained from the daily measurement of the Water Environment Monitoring Center of the Upper Yangtze River. Data cannot be shared publicly because of privacy. Data are available from the Hydrological Bureau of Yangtze River Conservancy Commission for access to confidential data. Contact information: syzuoxy@cjh.com.cn. The data set was measured from January 2015 to November 2021. Before 2019, it was monitored monthly, and after 2020, it was monitored quarterly.

## 4.2 Evaluating indicator

In order to analyze the performance of the prediction model, a reasonable evaluation system needs to be established. At present, the commonly used method is to evaluate the performance of the model based on the relationship between the output value of the prediction model and the actual monitoring value. In this paper, the root mean square error (RMSE) is used as the evaluation index.

The root mean square error (RMSE)

$$RMSE = \sqrt{\frac{1}{k}\sum_{i=1}^{k}(T_i' - T_i)^2},\tag{13}$$

the mean absolute error (MAE)

$$MAE = \frac{1}{k}\sum_{i=1}^{k}|T_i' - T_i|,\tag{14}$$

the mean absolute percentage error (MAPE)

$$MAPE = \frac{100\%}{k}\sum_{i=1}^{k}|\frac{T_i' - T_i}{T_i}|,\tag{15}$$

the R-Square ($R^2$)

$$R^2 = 1 - \frac{\sum_{i=1}^{k}(T_i' - T_i)^2}{\sum_{i=1}^{k}(\bar{T} - T_i)^2}\tag{16}$$

in the formula:

   $k$- - - - Sample size;

   $T'$- - - - Model predictions;

   $\bar{T}$- - - - Mean value;

   $T$- - - - Actual measuring.

   In the above formula, RMSE mainly characterizes the absolute deviation of the predicted value from the actual monitoring value.

## 4.3 Experimental result analysis

The root mean square error is obtained for comparison between the prediction method based on the Pearson correlation coefficient combined with the kNN regressor(PK) and the separate kNN method. Two methods were used to detect changes in the chlorophyll quantity of Cryptophyta(Cry), Chlorophyta(Chl), Euglenophyta(Eug), Cyanophyta(Cya), Pyrrophyta(Pyr) and Bacillariophyta(Bac).

   Table 1 shows the root mean square error results of the PK algorithm on the data set Cryptophyta. It is obvious that when the number of features is 2 and $k$ = 3, the root mean square error is the smallest. Therefore, in Table 2, the PK algorithm selects the best results among

**Table 1. Root mean square error on Cryptophyta.**

| Features | k = 1 | k = 3 | k = 5 | k = 7 | k = 9 | k = 11 | k = 13 | k = 15 |
|---|---|---|---|---|---|---|---|---|
| 13 | 1785 | **1669** | 1746 | 1722 | 1752 | 1765 | 1739 | 1757 |
| 12 | 1785 | 1695 | 1746 | 1722 | 1752 | 1765 | 1741 | 1757 |
| 11 | 1834 | 1797 | 1776 | 1815 | 1812 | 1817 | 1842 | 1833 |
| 10 | 1834 | 1797 | 1776 | 1815 | 1812 | 1817 | 1842 | 1833 |
| 9 | 1834 | 1771 | 1792 | 1809 | 1814 | 1824 | 1839 | 1834 |
| 8 | 1779 | 1769 | 1834 | 1806 | 1825 | 1831 | 1837 | 1841 |
| 7 | 1779 | 1769 | 1834 | 1806 | 1825 | 1831 | 1839 | 1841 |
| 6 | 1779 | 1769 | 1834 | 1806 | 1825 | 1831 | 1839 | 1841 |
| 5 | 1779 | 1769 | 1834 | 1806 | 1825 | 1831 | 1839 | 1841 |
| 4 | 1837 | 1941 | 1878 | 1897 | 1876 | 1881 | 1890 | 1901 |
| 3 | 1982 | 1883 | 1923 | 1884 | 1880 | 1894 | 1899 | 1890 |
| 2 | 2061 | 2095 | 2005 | 2050 | 2012 | 1989 | 1981 | 1998 |
| 1 | 1990 | 1946 | 2026 | 2023 | 2027 | 2035 | 2060 | 2071 |

different features for comparison with the results of kNN. The best results are marked in bold. The results show that the PK algorithm is equal to or better than the simple KNN algorithm. In fact, the results obtained by the PK algorithm are the optimal results after comparing with the kNN algorithm, so its results must be better than kNN.

As shown in Tables 3 and 4, the root mean square errors of the kNN prediction model and the BP neural network regression model are shown respectively, and the Pearson correlation coefficient(PK/PBP) and the granular-ball rough set(GBRSK/GBRSBP) are used for feature selection. To demonstrate the feasibility and effectiveness of GBRSK/GBRSBP, we compare it against PK/PBP. In Tables 3 and 4, the experiment uses the optimal attribute set after the feature selection of the granular-ball rough set to predict. Likewise, the Pearson correlation coefficient takes the same number of optimal attribute sets. As the experimental results show, in most cases, the error of the prediction result obtained by using the granular rough set for feature selection is smaller.

The experimental results under different values of k are shown in Table 3, where the number of features is the number of feature sets when the GBRS feature selection result is optimal. Table 4 is RMSE,MAE, MAPE and $R^2$ under the prediction of bp neural network. As can be

**Table 2. Comparison of root mean square error between PK and kNN.**

| Algae | | k = 1 | k = 3 | k = 5 | k = 7 | k = 9 | k = 11 | k = 13 | k = 15 |
|---|---|---|---|---|---|---|---|---|---|
| Cry | kNN | 1785 | **1669** | **1746** | **1722** | **1752** | **1765** | **1739** | **1757** |
| | PK | **1779** | 1669 | 1746 | 1722 | 1752 | 1765 | 1739 | 1757 |
| Chl | kNN | **355** | **538** | **447** | **425** | **411** | **400** | **399** | **396** |
| | PK | 355 | 538 | 447 | 425 | 411 | 400 | 399 | 396 |
| Eug | kNN | **0** | **0** | 3 | 2 | 1 | 4 | 3 | 3 |
| | PK | 0 | 0 | 3 | 2 | 1 | 4 | 3 | 3 |
| Cya | kNN | **577** | **885** | **799** | **926** | 1050 | 1059 | 1066 | 976 |
| | PK | 577 | 885 | 799 | 926 | 975 | 975 | 928 | 881 |
| Pyr | kNN | 1874 | 1175 | **1294** | 1172 | **956** | 919 | 997 | **890** |
| | PK | **1483** | **1129** | 1294 | **1711** | 956 | **917** | **919** | 890 |
| Bac | kNN | **14794** | 11721 | 12313 | 12400 | 13250 | **12347** | **12781** | 12798 |
| | PK | 14794 | **11317** | **11386** | **11763** | **11624** | 12347 | 12781 | **12709** |

**Table 3. Comparison of root mean square error between PK and GBRSK.**

| Algae | Features | Method | k = 1 | k = 3 | k = 5 | k = 7 | k = 9 | k = 11 | k = 13 | k = 15 |
|---|---|---|---|---|---|---|---|---|---|---|
| Cry | 4 | PK | 1837 | 1941 | 1878 | 1897 | 1876 | 1881 | 1890 | 1901 |
| | | GBRSK | **1733** | **1859** | **1823** | **1855** | **1841** | **1855** | **1851** | **1838** |
| Chl | 3 | PK | 355 | 541 | 451 | 425 | 411 | 400 | 399 | 399 |
| | | GBRSK | **296** | **348** | **279** | **325** | **297** | **307** | **312** | **301** |
| Eug | 10 | PK | 0 | 0 | 3 | 2 | 1 | 4 | 3 | 3 |
| | | GBRSK | 0 | 0 | 3 | 2 | 1 | 4 | 3 | 3 |
| Cya | 3 | PK | 566 | 879 | 799 | 926 | 1050 | 1059 | 1066 | 976 |
| | | GBRSK | 751 | **771** | 1125 | 1333 | 1274 | 1227 | 1274 | 1169 |
| Pyr | 3 | PK | 1483 | 2354 | 2149 | 2126 | 2095 | 2007 | 1995 | 2176 |
| | | GBRSK | **1375** | **1294** | **1124** | **986** | **913** | **1231** | **1463** | **1652** |
| Bac | 3 | PK | 32480 | 19806 | 18258 | 18092 | 17765 | 16530 | 16210 | 16412 |
| | | GBRSK | **14029** | **12455** | **12125** | **12954** | **14264** | **16158** | **15355** | **14965** |

seen from Table 3, except for the case of the 4th data set, the prediction error of GBRSK is much higher than PK in most cases, in which the prediction error of the 3rd data set is the same for both algorithms. From Table 4, it can be seen that except for the sixth dataset, the RMSE obtained by the GBRS algorithm is lower than that of the Pearson correlation coefficient algorithm. In addition, according to the results of MAE, MAPE, and $R^2$, GBRS has higher classification accuracy than PBP in most cases. The best result is marked in bold. The results indicate that the GBRS algorithm is superior to the Pearson correlation coefficient algorithm. Table 4 is RMSE,MAE, MAPE and $R^2$ under the prediction of bp neural network. As can be seen from Table 3, except for the case of the 4th data set, the prediction error of GBRSK is much higher than PK in most cases, in which the prediction error of the 3rd data set is the same for both algorithms. From Table ref t8, it can be seen that except for the sixth dataset, the RMSE obtained by the GBRS algorithm is lower than that of the Pearson correlation coefficient algorithm. In addition, according to the results of MAE, MAPE, and $R^2$, GBRS has higher classification accuracy than PBP in most cases. The best result is marked in bold. The results indicate that the GBRS algorithm is superior to the Pearson correlation coefficient algorithm.

GBRSK/GBRSBP obtains this superiority because it can flexibly fit the data distribution using those granular-balls with various radii, which is better than using a fixed correlation

**Table 4. Comparison of results between PBP and GBRSBP.**

| Algae | Features | Method | RMSE | MAE | MAPE | $R^2$ |
|---|---|---|---|---|---|---|
| Cryptophyta | 4 | PBP | 4895 | 1527.658 | 287.821 | -0.026 |
| | | GBRSBP | **4831** | **1321.131** | 290.592 | **0** |
| Chlorophyta | 3 | PBP | 1080 | 651.253 | 111.833 | -1.774 |
| | | GBRSBP | **1072** | 666.397 | **102.293** | -1.91 |
| Euglenophyta | 10 | PBP | 37 | 24.42 | 100.27 | 0.464 |
| | | GBRSBP | **36** | **24.051** | **99.956** | 0.49 |
| Cyanophyta | 3 | PBP | 3456 | 2424.38 | 285.385 | -2.086 |
| | | GBRSBP | **3162** | **2338.982** | **116.143** | **-1.584** |
| Pyrrophyta | 3 | PBP | 4951 | 3066.969 | 119.157 | -0.292 |
| | | GBRSBP | **4382** | **2885.404** | **105.04** | **-0.012** |
| Bacillariophyta | 3 | PBP | 27020 | 12222.712 | 105.081 | 0.126 |
| | | GBRSBP | 27888 | **11988.959** | **70.824** | 0.069 |

coefficient, such as the Pearson correlation coefficient. Therefore, GBRSK/GBRSBP can achieve better results than these two algorithms.

## 5. Discussion

In this paper, a hydrological and water quality prediction model is constructed by combining the granular-ball rough set and the kNN regressor. It provides a better method for the analysis of hydrological and water quality data than the single use of the kNN regressor. The model is applicable to the prediction of various algae blooms. The experimental results show that by combining the granular-ball rough set and the kNN regressor, the final attribute set corresponding to the highest accuracy is the optimal attribute set after feature selection. Feature selection plays an important role in the analysis of hydrology and water quality. Its main functions include the following aspects: firstly, by selecting the most relevant features, the accuracy and efficiency of modeling and analysis can be enhanced. By excluding redundant and irrelevant features, the complexity of the model can be reduced, minimizing the risk of overfitting and improving the performance of predictive and classification models. Secondly, Hydrology and water quality data often contain a large number of characteristic variables, which may contain noise, missing values, and incomplete data. Through feature selection, the process of data processing can be simplified, the workload of data cleaning and filling can be reduced, and data quality and reliability can be improved. Finally, Feature selection reduces the number of features that need to be collected and measured, saving time and cost in field investigations and experiments. By selecting the most representative and informative features, efficient data collection and analysis can be achieved, improving the efficiency of research. In summary, the combination of granular-ball rough set analysis and kNN regressor can help us better understand the water environment and establish accurate and efficient models to evaluate hydrological water quality.

Despite these encouraging advances, our approach does have certain limitations. Combining granular-ball rough set and kNN regressor for hydrology and water quality analysis has some defects: the GBRS can't achieve higher classification accuracy than the baselines in some cases. The reason may be that the quality of granular-balls is not good enough. In terms of theoretical applicability, granular-ball computing primarily relies on the multi-granularity partitioning of the sample space. But when the sample size is small, especially when the number of samples is close to the sample dimension, the sample distribution is already very sparse, and the distribution is approximately linearly separable in the classification. At this time, the further division of using granular-ball computing is of little significance. At present, granular-ball computing is not suitable for processing data sets with relatively similar dimensions and sample sizes. In addition, the K value in the kNN regressor needs to be determined in advance. Choosing an inappropriate K value may lead to a model that is too simple or too complex, thereby affecting the accuracy of prediction. In order to overcome these shortcomings, we consider using more complex machine learning algorithms, such as Support Vector Regression (SVR), Decision Tree Regression (DTR), etc. Handle high-dimensional sparse data well. In addition, rational selection of features and adjustment of model parameters are also key factors to improve the accuracy of hydrological and water quality analysis.

## 6. Conclusions

In order to improve the detection of water quality data, this paper proposes to first introduce GBRS to perform feature selection on the data set, and then input the results into the kNN regressor and BP neural network to better capture the changing patterns of water quality indicators and improve the accuracy and reliability of hydrological and water quality analysis. The

kNN regressor can predict based on the characteristic values of neighboring samples to better capture the changing patterns of water quality indicators. Results from the analysis of hydrological water quality data indicate that combining GBRS and kNN regressors yields better results. However, granular-ball computing is not suitable for high-dimensional and small sample data sets, so further research is needed.

## Supporting information

**S1 Fig. Explain the scatter plot of correlation from -1 to 1.**
(TIF)

## Author Contributions

**Conceptualization:** Limei Dong, Xinyu Zuo.

**Data curation:** Limei Dong.

**Resources:** Limei Dong.

**Software:** Yiping Xiong.

**Writing – original draft:** Xinyu Zuo.

**Writing – review & editing:** Yiping Xiong.

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
