## [Decision Letter · Decision Letter 0]

7 Dec 2023

PONE-D-23-30013Prediction of Hydrological and Water Quality Data Based on Granular-ball Rough Set and kNN AnalysisPLOS ONE

Dear Dr. Xiong,

Thank you for submitting your manuscript to PLOS ONE. After careful consideration, we feel that it has merit but does not fully meet PLOS ONE’s publication criteria as it currently stands. Therefore, we invite you to submit a revised version of the manuscript that addresses the points raised during the review process.

We look forward to receiving your revised manuscript.

Kind regards,

Dharmendra Kumar Meena

Academic Editor

PLOS ONE

Journal Requirements:

Additional Editor Comments:

The article can not be processed further, and its recommended for its major revision.

regards

D K Meena

Reviewers' comments:

Reviewer's Responses to Questions

**Comments to the Author**

1. Is the manuscript technically sound, and do the data support the conclusions?

Reviewer #1: Yes

Reviewer #2: Yes

Reviewer #3: Partly

Reviewer #4: Partly

Reviewer #5: Yes

Reviewer #6: Partly

2. Has the statistical analysis been performed appropriately and rigorously? 

Reviewer #1: Yes

Reviewer #2: Yes

Reviewer #3: Yes

Reviewer #4: Yes

Reviewer #5: Yes

Reviewer #6: Yes

3. Have the authors made all data underlying the findings in their manuscript fully available?

Reviewer #1: No

Reviewer #2: Yes

Reviewer #3: No

Reviewer #4: No

Reviewer #5: Yes

Reviewer #6: Yes

4. Is the manuscript presented in an intelligible fashion and written in standard English?

Reviewer #1: Yes

Reviewer #2: Yes

Reviewer #3: Yes

Reviewer #4: Yes

Reviewer #5: Yes

Reviewer #6: Yes

5. Review Comments to the Author

Reviewer #1: Overall great literature with novel contributions.

These are a few comments:

- Granular-ball Rough Set not clearly explained in the introduction.

- Repetition of “The basic idea of the kNN regression algorithm is to select the nearest training samples in K Euclidean spaces according to the similarity of Euclidean distance between samples, and the regression value is the mean or weighted value of K-nearest neighbor samples. ” under kNN regressor subsection

- BP neural network, what is BP? Could you make it more clear.

Reviewer #2: 1. The manuscript needs significant improvement in terms of language. A proofreading is required.

2. The research lacks novelty. The author needs to explain and emphasize more the innovation of this work in the introduction.

3. Starting of abstract seems not smooth, need major revision of abstract by including

brief methodology, results, and findings of the manuscript.

4. Introduction section was written briefly, please add more literature about the research methodology. Furthermore, a comparison should be made over the operating conditions and yields in the literature rather than general information. The original aspect of the study should be emphasized. In addition, the aim of the study should be stated in more detail.

5. The methodology was not based on reliable references.

6. More explanation is needed for where there is a research gap and what the goals of the research are. The research gap and the goals of the research are not explained in detail, which leads to the reader missing the significance of the research.

7. The authors should compare the results with those of the previous researches, and this comparison should be presented in an individual table.

8. Extend the conclusions with all your most important findings.

9. The references need to be checked to make sure they are complete and up to date.

Reviewer #3: Comment 1: Its better if the author follows the guideline of manuscript preparation of this journal (PLOSE ONE)

Comment 2: In abstract part, you didn’t state the objective of the research. It’s better if the statement of the problems, research gap and objective is stated very well. Also its better if you don’t use abbreviation in title and abstract part ‘kNN’

Comment 3: the study has no line number

Comment 4: in introduction part, first line its better if you cite after the end of the sentence.

Comment 5: the introduction part is very bulky. It needs to make it clear by modifying it. The research gap and objectives have to clearly provided in this part.

Comment 6: on page 3, the (equation…1) wasn’t cited. It needs to properly cite when you study scientific research.

Comment 7: in methodology part the research design have to properly stated. Therefore, it needs major revision and modification

Comment 8: on page 13, second paragraph discussion part, you didn’t discuss the limitation of your approach. It’s better if you clearly state the limitation on this study and also provide your recommendation.

Reviewer #4: The manuscript, titled "Prediction of Hydrological and Water Quality Data Based on Granular Ball Rough Set and kNN Analysis," demonstrates technical robustness, offering a comprehensive overview of the employed methodology, and incorporates a meticulous statistical analysis. To enhance its quality, a more detailed introduction is recommended, focussing on comparing various prediction approaches or models, as the current content emphasises algal bloom in the introduction section.

One notable gap in the paper is the lack of information regarding the method employed for data preprocessing. This crucial step in the research process should be elaborated on to provide transparency and enable replication by other researchers.

Furthermore, the manuscript lacks a specification regarding the duration of the data periods studied. Clarifying the temporal scope of the data is essential for a thorough understanding of the research context and findings.

Lastly, the absence of predicted time-series data for water quality is a notable limitation. Including this information would significantly enhance the completeness of the manuscript and contribute to a more holistic presentation of the study's outcomes.

Addressing these gaps, namely, providing a detailed introduction comparing prediction approaches, elucidating the data preprocessing method, specifying the duration of data periods, and incorporating predicted time-series data for water quality, will significantly improve the manuscript's overall quality and render it suitable for publication.

Reviewer #5: This paper proposes a methodology of synergizing Granular-ball rough set analysis with kNN regression to dissect hydrological water quality data. The paper is very well written, which enables to Improve the accuracy and reliability of hydrologic and water quality analysis.

1.In INTRODUCTION, the latest relevant research background needs to be supplemented. The research background of the data analysis method and regression analysis algorithm used in this paper in hydrology and water quality is rarely introduced, and there is also a lack of new literature. How the study relates to this previously published research and how the model approach you used differs from previous research?

2.Figure 2, Figure 3, and Figure 4 all show a flow cahrt of the method usd in this paper, and authors should consider a more comprehensive and concise presentation.

3.From Table 2 to Table 8, the authors use a total of eight tables to show the mroot mean square error, suggesting a more concise display method or a choice.

4.Table 7 and Table 8 respectively introduce the root mean square error results using KNN regression model and BP neural network, but the content formats of table header are not uniform.

5.Most of the table formats need to be adjusted, for example, the header lines in Tables 1 to 7 are too high.

6.Is it limited to use only root mean square error as an evaluation index to predict model performance? Why not use more evaluation indexes for comprehensive analysis?

7.This paper only mentions the sources of water level, flow, water quality and ecological data, which should be supplemented with specific information on hydrology and water quality.

8.Two different algorithms, KNN regressor and BP neural network, and two data analysis methods, Granular-ball Rough Set analysis and Pearson correlation coefficient, used in this paper have been widely used in many fields. In this paper, different algorithms and data analysis methods are combined respectively to analyze, and the main conclusion is that the combination of Granular-ball Rough Set analysis and kNN regressor better results than using independent kNN regressor. First of all, the results of independent use of kNN regressors are not included in this paper. Secondly, simultaneous innovation has not been explained and highlighted in detail. Whether the conclusions are universal, and whether the data results are different in different places or time periods.

9.The format of the references is inconsistent, such as the 45th reference and the 53rd to 59th reference.

This paper is well written, but there are still some shortcomings, so the recommendation given is major revision.

Reviewer #6: Comments to author

1. The abstract needs revision. It is always better to provide quantitative results in the abstract. The conclusion mentioned in abstract is too generic 'superior result' rather than in what term prroposed approach was superior to KNN would be meaningful.

2. Figure 1. I don't see the essence of Figure 1. It can be moved to annex or supplementary section.

3. Page 7: In both the subsection you have mentioned about the proposed flow chart in this article. However, the title shows that your proposed methodology is granular ball and KNN regression. Please justify.

4. The paper has too many tables. Try to reduce to 4 numbers rather than 8

5. I am not sure whether the authors are clear on the works they are doing. Somewhere they have mentioned about hydrology and water quality and somewhere hydrology water quality (in abstract). Be consistent on the work you are doing. What difference do the author want to portray from these discrepancies, if there are any.

6. PLOS authors have the option to publish the peer review history of their article (what does this mean?). If published, this will include your full peer review and any attached files.

Reviewer #1: **Yes: **Daniel Agyapong

Reviewer #2: No

Reviewer #3: **Yes: **Siraj Abduro Abdulahi

Reviewer #4: No

Reviewer #5: No

Reviewer #6: No

---

## [Author Response · Author response to Decision Letter 0]

4 Jan 2024

Specific responses can be seen in Response to Reviewers.docx

---

## [Editor Report · Decision Letter 1]

9 Jan 2024

PONE-D-23-30013R1Prediction of Hydrological and Water Quality Data Based on Granular-ball Rough Set and k-nearest neighbor AnalysisPLOS ONE

Dear Dr. Xiong,

Thank you for submitting your manuscript to PLOS ONE. After careful consideration, we feel that it has merit but does not fully meet PLOS ONE’s publication criteria as it currently stands. Therefore, we invite you to submit a revised version of the manuscript that addresses the points raised during the review process.

We look forward to receiving your revised manuscript.

Kind regards,

Dharmendra Kumar Meena

Academic Editor

PLOS ONE

Additional Editor Comments:

author are advised to provide the response to Reviewer doc file.

---

## [Author Response · Author response to Decision Letter 1]

23 Jan 2024

Specific responses can be viewed Response to Reviewers.docx.

---

## [Editor Report · Decision Letter 2]

30 Jan 2024

Prediction of Hydrological and Water Quality Data Based on Granular-ball Rough Set and k-nearest neighbor Analysis

PONE-D-23-30013R2

Dear Dr. Xiong

We’re pleased to inform you that your manuscript has been judged scientifically suitable for publication and will be formally accepted for publication once it meets all outstanding technical requirements.

Kind regards,

Dharmendra Kumar Meena

Academic Editor

PLOS ONE

Additional Editor Comments (optional):

The article can be accepted
---

## [Editor Report · Acceptance letter]

13 Feb 2024

PONE-D-23-30013R2 

PLOS ONE

Dear Dr. Xiong, 

I'm pleased to inform you that your manuscript has been deemed suitable for publication in PLOS ONE. Congratulations! Your manuscript is now being handed over to our production team.

Kind regards, 

on behalf of

Dr. Dharmendra Kumar Meena 

Academic Editor

PLOS ONE